# Introduction to Dione's Wispy Terrain as a Putative Model Region for "Micro" Wilson Cycles on Icy Satellites

**Balázs Bradák** [1,*], **Jun Kimura** [2], **Daisuke Asahina** [3], **Mayssa El Yazidi** [4] and **Csilla Orgel** [5]

1   Laboratory of Exo-Oceans, Faculty of Oceanology, Kobe University, 5-1-1 Fukaeminami-machi, Higashinada-ku, Kobe 658-0022, Japan
2   Department of Earth and Space Science, Osaka University, 1-1 Machikaneyama-cho, Toyonaka 560-0043, Japan
3   Research Institute of Earthquake and Volcano Geology, Geological Survey of Japan/AIST, 1-1-1 Higashi, Tsukuba 305-8560, Ibaraki, Japan
4   European Space Agency, European Space Research and Technology Centre (ESA/ESTEC), Keplerlaan 1, 2201 AZ Noordwijk, The Netherlands
5   Directorate of Human and Robotic Exploration, European Space Agency, European Space Research and Technology Centre (ESA/ESTEC), Keplerlaan 1, 2201 AZ Noordwijk, The Netherlands
*   Correspondence: bradak.b@port.kobe-u.ac.jp

**Abstract:** The Wispy Terrain is the region of chasmata characterized by quasi-parallel fault systems, formed by extensional and shear stresses of the icy crust of Dione, a moon of Saturn. Besides the basic, satellite-scale geological mapping and very general definition of the phenomenon, only a few studies focus on the Wispy Terrain and its chasmata from the angle of detailed tectonic reconstruction, with others mainly targeting, e.g., the timing of its formation. This study provides a detailed geological and cryotectonic analysis in the surroundings of the Eurotas and Palatine Chasmata and proposes additional, until now, unidentified tectonic processes and a formation model. The relationship between fragmentary impact craters and tectonic features indicates other newly suspected tectonic movements, namely thrust, and splay and décollement fault systems. In contrast to the commonly expected and identified dilatational processes, such fault types show compression and are characteristic of subduction in a terrestrial environment. Theoretically, the appearance of such tectonic processes means that the already-known rift and the newly discovered subsumption (subduction-like) processes may appear together in the Wispy Terrain. The appearance of both features may suggest the presence of some of the components (phases) of a Wilson cycle analog cryotectonic cycle (or possibly cycles) in icy planetary bodies like Dione.

**Keywords:** Wispy Terrain; Dione; Saturn; icy satellite; subsumption; Wilson cycles; resurfacing



## 1. Introduction

### 1.1. The Wispy Terrain

The icy moon Dione (a satellite of Saturn) and its enigmatic feature, the so-called Wispy Terrain, have been the target of a long-lasting scientific debate and ongoing research. The region was named Wispy Terrain after observing frequently appearing wispy streaks, markings, and lineaments in the images of the Voyager spacecraft. The higher resolution images of the Cassini spacecraft revealed that the markings of Wispy Terrain and the so-called chasmata or chasm system consist of quasi-parallel faults, troughs, and more complex "horst and graben" structures indicating extensional and shear stresses in Dione's icy crust. Despite the popularity of Wispy Terrain as a research target, there are still ongoing debates about the evolution of the highly tectonized region of the icy satellite. Neither the age of the surface nor the formation of various features observed in the area are fully understood.

The early planetary geologic mapping of Dione provided a preliminary view of the features of the Wispy Terrain. Still, beyond a chronostratigraphic proposal, no precise age determination could be made about its formation time [1,2]. Following the analysis of

the more detailed Cassini images, the impact crater dating from the Eurotas Chasmata (Figure 1) indicated that the surface of Wispy Terrain is relatively young and the cryotectonic processes in the region might have been active until ~3 Ga or even until ~1 Ga ago, keeping the uncertainty in the cratering models in mind [3,4]. Almost three decades passed after the first geological mapping, and the formation of the main terrains of Dione was re-evaluated in light of the new results [5]. The chasmata systems, including the Eurotas and Palatine Chasmata (Wispy Terrain), were defined as fractured cratered plains and subdivided into three facies types, regarding the timing of their formation by tectonic episodes, which date back to 3.7 Ga (with 100 Ma uncertainties) or between 2.7 Ga and 260 Ma [5]. Later calculations showed 4.5 (+0.2/−2.7) and 2.5 (+2.0/−1.9) Ga computed impact crater ages for the Faulted Terrain, including the Wispy Terrain. The results suggested that those ages do not reflect the timing of cryotectonic activity, but the period while the larger craters and their ejecta blankets erased and covered the smaller craters in the region [6]. Among the newer studies, Fergusson et al. [7] suggest that surface renewal is not only limited to certain areas but appears in many regions on Dione.

Along with the continuously updated impact crater ages, there has been some development in theories related to the formation of the region as well. Based on the analog geological characteristics between the fault system of the chasmata and Earth's provinces with divergent tectonic plates, Dione's Faulted Terrain (along with the Rhea and Tethys chasmata) was defined as hemisphere-scale rift zones [8,9].

In addition, the possibility of a subsurface ocean under the satellite's ice shell brought a promise of a still active surface and a potential for life under the icy cover [10]. The study of the stratigraphic relationship between the craters and faults on the Wispy Terrain suggests that the faulting is a geologically very recent event, dating back to 0.3–0.79 Ga [11]. Some of the newest studies go even further and suggest that the upper limit for the age of the studied fault on Dione's Wispy Terrain is only 152 Ma, which supports the hypothesis that the cryotectonism might be still active or was active a very short time ago (ca. 100 Ma) on the satellite [12].

One of the primary research topics about icy satellites with potential subsurface oceans (e.g., Europa) is the understanding of downward processes from the frozen surface and the description of the material exchange between surface and subsurface regions, which may markedly contribute to the oxygenation of the subsurface oceans and provide potential habitat for living organisms. Ice or cryotectonic processes may support such downward material transport [13].

*1.2. A Brief Review of Dione's Cryotectonic Features*

During the more and more detailed study of the satellite, various cryotectonic features have been observed in the icy crust of Dione, which are summarized in Table 1. Out of the different basic fault types (normal, reverse, and strike-slip faults), normal faults seem to be the most common, indicating an extensional stress field and resulting in the formation of various simple and more complex tectonic features, such as troughs, scarps, and the so-called horst and graben structure [2,4,14,15] (Table 1).

Linear features have been named in various ways since the first images from Dione arrived, such as "bright wispy markings" [1,2,14], "bright lineaments" [4,14], in general, "lineaments" [2], fossae, and chasmata (chasma) [2,16]. The terms fossae and chasmata have been applied to linear depressions with various depths. Still, such features have many characteristic marks, which may indicate different ways of formation and maturity in their development, e.g., on a scale from "simple" troughs to complex ones, even with parallel ridges in their bottom [1,2,16]. They often appear as high albedo features, most likely due to (i) the exposure of ice on the slopes of the scarps in the case of lineaments with a minimal vertical displacement [4,14], (ii) some bright material coating on the wall of some troughs (if the linear feature is identified as a trough), (iii) the lineaments had some Sun-facing slopes when the image was taken [1,2], and (iv) some subsurface material, which might resurface during various (cryotectonic–cryovolcanic) processes [1,2]. In the case of Eurotas,

Palatine, and Padua Chasmata, the structures reached a specific size and complexity, which led to further theories describing the system of scarps, troughs, and normal faults as rift zones, similar to the ones on other satellites, such as Rhea and Tethys [8,9].

**Table 1.** The summary of putative cryotectonic features observed on the surface of Dione.

| Putative Cryotectonic Features | Short Description | Stress Field | References |
|---|---|---|---|
| Bright wispy markings | (a) Surficial deposit of high-albedo material associated with eruptive events along fractures<br>(b) Trough walls coated by bright material<br>(c) Lineaments with slopes facing toward the Sun<br>(d) Fault scarps | Extension (divergence) | [1,2,14] |
| Fossae ǀ Chasmata | Initially, such features were called "wispy material" or "wispy markings" (former lineae) | Extension (divergence) | [16] |
| Fractures | Non-Wispy Terrain features (from polygonal impact craters); fractures consistent with the global deformation from a combination of satellite despinning and volume expansion | | [15] |
| Lineaments | In general, various linear features; characteristic global lineament trend (NE and NW—middle latitudes and equatorial region; E–W—polar region) → origin; stresses by (i) loss of angular momentum associated with despinning, and (ii) effect of orbital recession, superimposed on (iii) tensional stress (global surface extension) | Extension (divergence) | [2] |
| (Bright) lineaments | Widely abundant; single or densely spaced, sub-parallel lineaments; may reach lengths of several hundred kilometers; some of them may be scarps with slight vertical displacement | Ext. (diverg.) or compr. (converg.) | [4,14] |
| (Radial) lineaments or ray crater | Cassandra: bright ray crater/system of radial lineaments and scarps (set of radial scarps radiating away from a point source); bright exposure of ice on the slopes of the scarps | Extension due to diapir formation | [4,14] |
| Ridges (Janiculum Dorsa) | Flexural deformation: 500 km long, north–south trending ridge; flanked by parallel flexural depressions; leading hemispheres; 4 Ga old | Compression (convergence) | [17] |
| Ridges | Extending 50–300 km long, <0.5 km high, broad, "convex in cross-section" features; merge into lineaments or escarpments; parallel and subparallel ridge systems → origin ((volcanic or) tectonic): (a) parallel normal faults; (b) fault scarps; (c) graben; (d) high-angle reverse faulting<br>Ridge complex: prominent ridge associated with minor sub-parallel ridges and troughs (Janiculum Dorsa) | Extension (divergence; a, b, and c), or compression (convergence; d) | [2,4] |
| Rift zones | Large-scale extensional deformation (extends ~1300 km, subtend 133° of arc, and varies 40–130 km in width); concentrated within or at the borders of the trailing hemisphere; shows a preference for ~N–S-oriented strikes (Palatine, Eurotas, and Padua Chasmata); complex fault structure | Extension (divergence) | [8,9] |
| Scarps | Extending ≤100 km long, <1 km high, linear, uncommon features; extension of large polygonal crater rims → origin: (a) faulting; (b) mass-movements | | [2] |
| Broad bands | Formed by densely spaced (graben) horsts and scarps; bright albedo due to exposure to clean water ice | Extension (divergence) | [4] |
| Shallow normal fault slopes | Normal faults with steep dips → viscous relaxation (due to lithospheric heating events related to radionuclide decay), which affects fault slopes (Padua and Palatine Chasmata) | Extension (divergence) + viscous rel. | [15] |
| Troughs ǀ chasma ǀ chasmata | Long linear, narrow, or wider (shallower) troughs; branching (Tibur Chasma and Larissa and Latium Chasma); parallel ridges may appear in their bottom (Tibur Chasma); ~30 to 100 km (>500 km), 0.3 ± 0.1 km deep, irregular, or scalloped walls, rims may appear; rectilinear troughs → grabens; parallel rectilinear troughs → horst and graben structure (Palatine Chasmata) | Extension (divergence) | [1,2] |
| Troughs | Several kilometers or a few tens of kilometers wide; several hundred kilometers long; linear, arcuate, or curved; single or in parallel sets | Extension (divergence) | [4] |

Most of the studies involved in the study of various tectonic features agree about the formation process of the morphological features described above, including the overall extension- (and shear stress-) dominated stress fields in the icy crust of the satellite (the triggering processes will be discussed later in this section). Such agreement does not apply to ridges, which are interpreted as (i) parallel normal faults, (ii) fault scarps, (iii) graben, (iv) high-angle reverse faults [2], and (v) flexural deformations [17]. Sup-

pose one of the former three applies to ridges. In that case, the stress field during its formation agrees with the stress field that appears during the formation of troughs, chasmata, and other features suggested above. Suppose the ridges are defined as reverse faults and flexural deformations (the latter two). In that case, it indicates compression (or shear stress), which would be unusual regarding the studies about cryotectonic features on the surface of Dione (Table 1).

In general, in his early synthesis of the geological evolution of Dione, Moore [2] suggested sequences of heat, expansion, melting, and contraction as a trigger of stresses in its icy crust. Regarding the processes that may induce those stresses and result in the formation of various cryotectonic deformation-related features, various mechanisms, such as changes in volume due to phase changes within a satellite [18], solid-state convection [19], polar wander [20], diurnal tidal flexing [21,22], forced eccentricity [22,23], and obliquity [22,24–26], the nonsynchronous rotation of the ice shell to the tidal torques [27,28] can be mentioned. Along such long-lasting (at a geological scale), quasi-permanent, and periodic processes, catastrophic events, namely large impacts, may be considered the triggers of deep-seated lines of weakness in the crust that later determined the location of cryotectonic features [2,29].

In some cases, coincidental findings turn research in new directions. The revisit and study of the tectonic features found on Dione's Wispy Terrain is one of those cases. The detailed crater mapping on the satellite, partly introduced in Bradák et al. [30], revealed the frequent appearance of fragmentary or incomplete craters in the region located between Palatine Chasmata and Eurotas Chasmata, which may indicate compression in specific locations instead of the highly expected dilatation (please find a detailed explanation in Section 2.2). Using craters as stress field indicators may reveal additional information about one of the most enigmatic features of Dione's surface, the chasmata system of the Wispy Terrain. This study aims to identify various tectonic features and stress fields by a crater-morphology-based method and reconstruct some phases of Wispy Terrain's tectonic evolution, one of the most and still enigmatic regions of Dione.

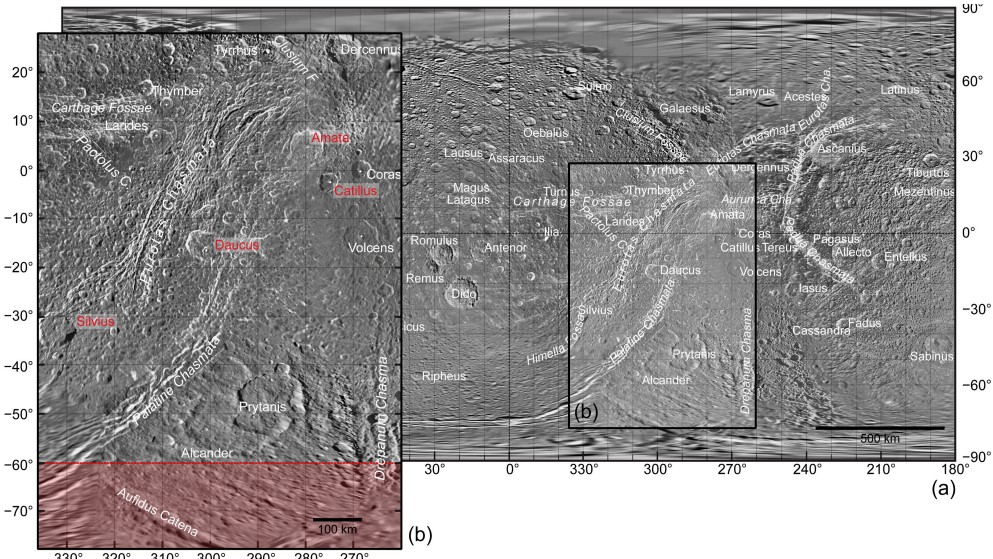

**Figure 1.** (**a**) The Cassini–Voyager Global Mosaic 154m v1 map, with (**b**) the study region in the Wispy Terrain of Dione. The Cassini–Voyager Global Mosaic 154 m v1 map can be found at https://astrogeology.usgs.gov/search/map/Dione/Voyager/Dione_Cassini_Voyager_mosaic_global_154m (accessed on 28 February 2023); reprinted/adapted with permission from the study of Batson 1984 [31], Greeley and Batson 2007 [32], Roatsch et al. 2016 [33], and Schenk 2016 [34]. Dashed red lines at latitude –60° and the transparent red area below, southward, indicate the region with mapping uncertainties due to the projection-related distortion of the surface features. The craters, denoted in red, are analyzed later in detail (Sections 3 and 4).

## 2. Materials and Methods

### 2.1. Data

The studied region, the so-called Wispy Terrains, extends approximately between latitude 30° and −80° and longitude ca. 260° and 340° (westward) (Figure 1). The target area of this research is the region located between Palatine Chasmata and the "southern" section of Eurotas Chasmata. The base of the geological and tectonic mapping is the Cassini–Voyager Global Mosaic 154m v1 map, which can be found at the Astropedia, Lunar and Planetary Cartographic Catalog internet site [31–34]. The applied nomenclature follows the recommendation of the Gazetteer of Planetary Nomenclature [35].

The geological mapping and some related GIS research were performed by QGIS 3.22 software, followed by statistical analysis, which was executed by various Python 3.10.4 software packages, including NumPy 1.22.012, matplotlib, pandas, SciPy 1.8.0, and seaborn. All features, craters, and structural elements were interpreted visually and digitized interactively from the high-resolution and georeferenced images in QGIS 3.22. Basic information about the craters and lineaments was collected in a QGIS database and was used as a primary source for the analysis in Python 3.10.4 software. The Python code for the rose diagram analysis, used during the reconstruction of cryotectonic stress fields, was created by De Pinho [36].

During the mapping, various crater types (for details, please see Section 2.2) and lineaments were identified. The mapping of craters and linear features also had some limitations in certain areas due to the distortion of the map (e.g., at higher latitudes).

Some mapping uncertainty also needs to be addressed here, related to the low resolution of certain parts of the Cassini–Voyager Global Mosaic map. This might influence the number of identified craters and lineaments, making clear identification of their types difficult or impossible.

Please note that the term Wispy Terrain in the study refers to the area between and along Eurotas and Palatine Chasmata. Although the term linea was used for certain geological features on Dione, this study used the terms line, linea, and lineaments for any linear features in general.

### 2.2. A Potential Tool for Stress Field Reconstruction: Crater Preservation in Various Cryotectonic Settings

Special attention was paid to the appearance of the so-called polygonal craters and "crater fragments" during the detailed mapping of the region. Polygonal craters and/or crater fragments are craters with their floor or wall cut by faults, which might cause the detachment of certain parts and their vertical or horizontal allocation. In some studies, crater fragments, or specific types of crater fragments, are called polygonal craters and are defined as indicators of fracturing within planetary bodies' rocky and icy shells [37,38]. Polygonal craters are described as having at least two connected straight rim segments, and their polygonal shapes are formed by jointing triggered by underlying faults [37,39]. Polygonal craters are observed in many regions of Dione [38] and can provide information about various cryotectonic periods forming and reforming its ice shell [7]. During the analysis of the Wispy Terrain, the following crater types were targeted. Please note that the following classification of the craters only focuses on possible tectonic deformations and not any other morphological features, such as lobes and peaks.

Type P ("poly" or polygonal): In some cases of short-distance horizontal allocation of the crater fragments (strike-slip faults), the cryotectonic activity "only" distorts the crater rim, resulting in visible sharp edges in the supposedly curved rim. The whole polygonal crater is still recognizable.

Type SSF (strike-slip fault): In some instances of horizontal allocation (strike-slip faults), the parts of the crater may move further distances, but even in such cases, the parts are recognizable along the fault trace, and the crater is reconstructible.

Type NRF (normal or reverse fault): Normal (or reverse) faulting may cause the allocation of the crater fragments along a fault plane with a certain dip to the horizontal plane. Based on the angle of the dip (deep to shallow) and the orientation of the fault planes

of individual blocks in a fault system, various fault types may form on the crust of icy satellites, such as normal faults, horst and graben structures, domino-style fault blocks, and listric normal faults [15]. Among many similarities, one is essential regarding preserving the crater fragments separated by deep- or shallow-angle faulting. Despite the allocation of the crater parts, the fragments are recognizable on the original surface of the footwall and hanging wall block, and the "whole" crater is reconstructible.

Type TF (thrust fault): The commonality in the suggested processes above is that, in every case, the parts of the crater are recognizable and the crater itself is reconstructible. One of the exceptional cases would be the formation of thrust faults, a reverse fault with a shallow dip, with a hanging wall that moves up and over the footwall. In such cases, the hanging wall may "cover" a certain part of the footwall top, resulting in fragmentary or incomplete craters ("half-craters") without any recognizable craters and without completing other connecting crater parts in their surroundings.

The key (and simplified) difference between the faults is that various types of normal faults are formed in extensional and thrust faults (low-angle reverse faults) in a compressional stress field, and the displacement of the thrust fault is quasi-parallel to the compression force. A strike-slip fault can be deformed by both compression and tension forces, and the direction of a strike-slip fault is about 30–60 degrees to the compression or tension force.

During the mapping of the craters, the following types were categorized: (i) Type NT—any craters without the mark of tectonic deformation, (ii) Type P—polygonal craters formed by various tectonic processes (Figure 2a), (iii) Type SSF—craters with various degrees of detachment, as the possible result of strike-slip faulting (Figure 2c), and (iv) Type TF—fragmented or "incomplete" craters, which may indicate thrust fault cryotectonic settings formed in a compressional stress field (Figure 2d).

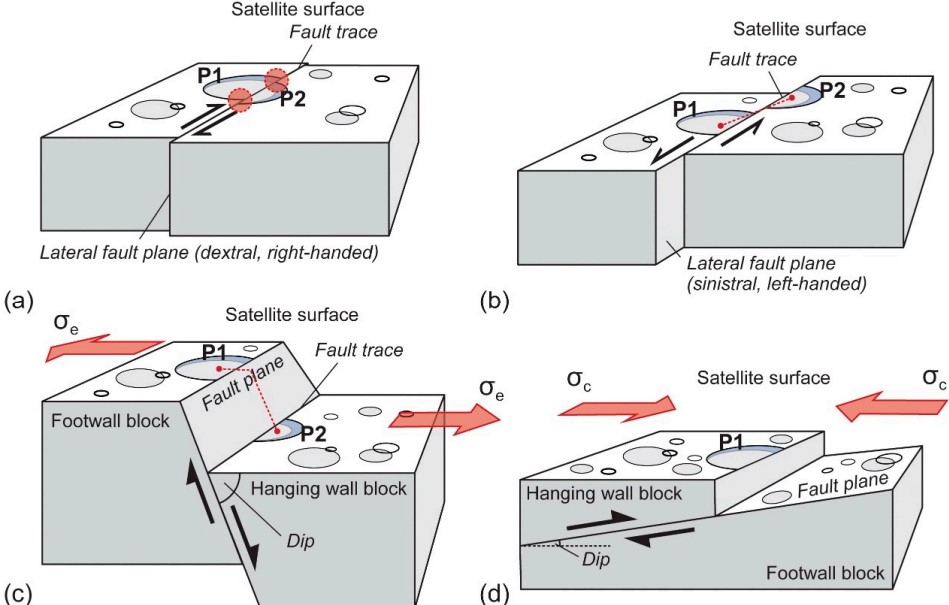

**Figure 2.** Crater preservation in various fault settings, a simplified view. (**a**) Type P: strike-slip fault with minimal horizontal re-location; (**b**) Type SSF: strike-slip fault with the dislocation of crater parts; (**c**) Type NRF: normal fault with vertical re-location of some crater parts; and (**d**) Type TF: formation of fragmentary craters through thrust faulting. Special attention was paid to the appearance of crater fragments during the detailed mapping of the region. Polygonal craters and crater fragments are craters with their floor or wall cut by faults, which might cause the detachment of certain parts and their vertical or horizontal allocation. Such processes result in the deformation of the initially circle- or oval-shaped crater rim, transforming it into a polygonal shape. (Please find more information in the text, Section 2.2). Circles in Figure 2a indicates the slight displacement of the separated crater

parts; red dashed lines with dots at the ends indicates the identifiable connections between the allocated crater parts; Red arrows indicates the stress fields and their nature, $\sigma_e$, and $\sigma_c$, standing for extension and compression respectively.

This study focused on "simple" crater types, and although they may provide additional information, the distribution of multiple or hybrid crater types was not analyzed.

Due to the possible bias related to the lack of clear identification of the connection between the allocation of crater fragments and vertical movements, Type NRF craters were not analyzed. To resolve the lack of Type NRF craters in the analysis of tectonic features, along with the polygonal crater morphology and the allocation of the crater fragments, possible tectonic features appearing in the walls of larger craters were analyzed. Despite resolution-related issues, which must be considered during the interpretation of the observations, in some cases, both the surface (horizontal) and vertical components of the appearing tectonic feature could be recognized.

## 3. Results

### 3.1. The Distribution of Various "Indicator" Crater Types

During the geological mapping of the region, close to seven thousand craters were mapped. A total of 94.2% of craters belong to non-tectonized, Type NT craters. Less than 1% of the craters are identified as Type P and Type SST craters, 0.7 and 0.6%, respectively. A total of 4.5% of the craters are defined as Type TF crater fragments, possibly indicating thrust faults.

The size of the craters is 0.9 to 98.9 km diameter in the case of Type NT and 0.8 to 61 km diameter in the case of Type P craters. In the case of Type SSF craters, the diameter of the craters was derived from their area, falling between 3.5 and 59.3 km. The derived diameter might be slightly overestimated, resulting in diameters between 1.1 and 132.4 km and 1.9 and 67.5 km for Type NT and Type P craters, respectively. Additional crater statistics provide more detailed information. A total of 75% of the Type NT craters have smaller diameters than 4.9 km (quartile—Q3: 4.9 km). The mapped Type P craters are generally more prominent in diameter than Type NT craters. A total of 25% of the crater diameters are below 4.1 km (Q1), 50% are below 6.95 km (Q2 or median), and 75% are below 13 km (Q3). Such differences may appear because it is easier to recognize polygonal craters at bigger crater sizes. Observing the image at particular magnifications, the smaller, non-tectonized, and polygonal craters may look identical due to the low resolution and pixelization. This rule may apply to Type SSF craters as well. It is increasingly challenging to identify and reconstruct a whole crater by putting it together fragment by fragment, appearing along a tectonic line when those fragments are smaller and smaller.

In the case of the incomplete Type TF craters, the area of the identified fragments was calculated and studied. Similar to all types, the maximum area size indicated an extreme value, 2083 km$^2$, compared to other statistical parameters, such as the average and median of the data, 57 and 22 km$^2$, respectively. The study of the quartiles suggests a similar crater area distribution to other types, with Q1: 11.4 km$^2$, Q2: 22 km$^2$, and Q3: 51.4 km$^2$, respectively. As is expected, most of the craters, supposedly affected by cryotectonic processes, are located along the lineament, especially along Eurotas and Palatine Chasmata (Figure 3).

Basic statistics provide general information, group by group. Still, understandably, a deeper comparison of the diameter and area of the whole craters (Types NT and P), reconstructed full craters (Type SST), and incomplete ones (Type TF) may be unnecessary and can be misleading.

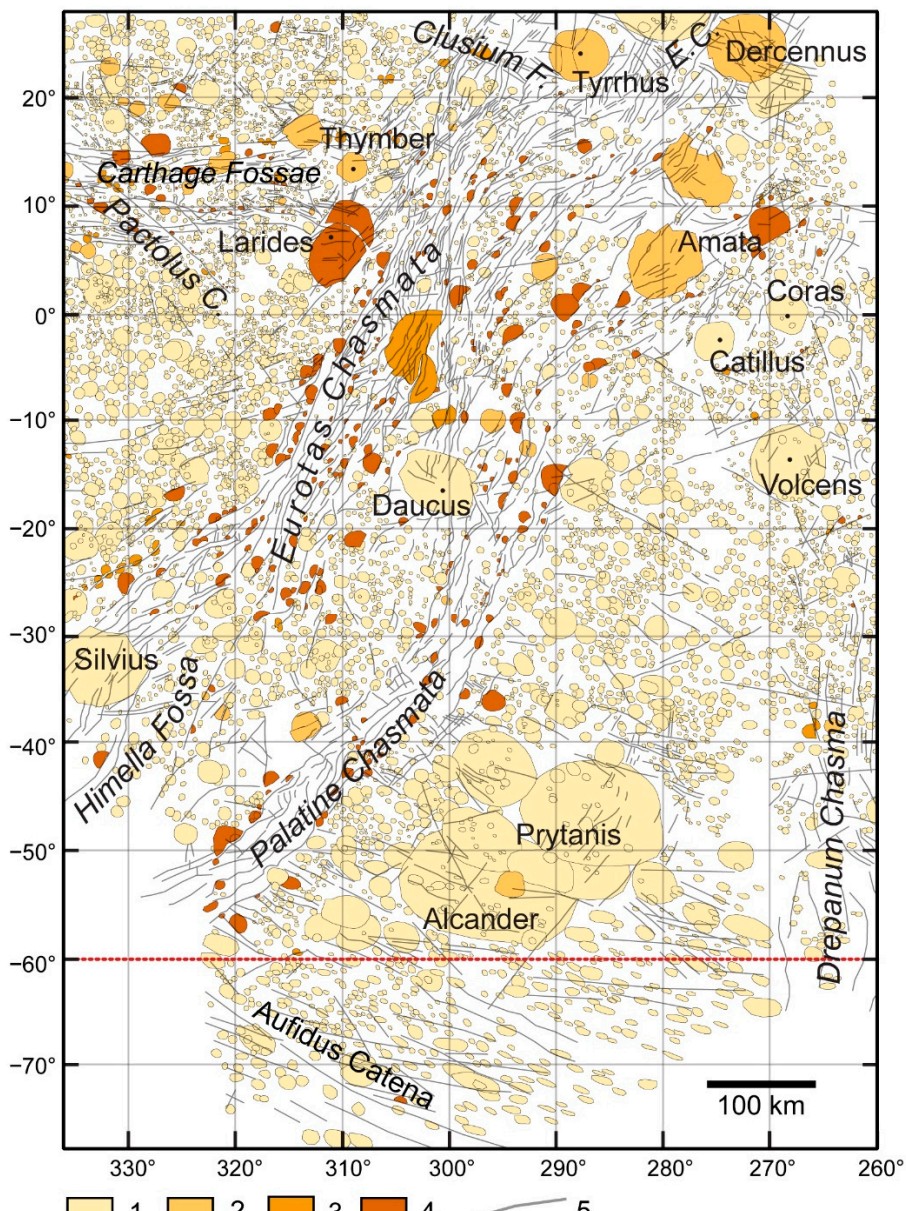

**Figure 3.** The primary geological map of the studied area of the Wispy Terrain focuses on polygonal and fragmented craters. The relationship between various tectonic features is based on the distribution of polygonal craters and crater fragments and the analysis of large crater wall "outcrops". The marks with different numbers indicate 1—"normal", non-tectonized craters; 2—Type P craters; 3—Type SSF craters; 4—Type TF craters; and 5—suspected tectonic movements. The dashed red line at latitude −60° indicates the region's boundary, with mapping uncertainties southward due to the projection-related distortion of the surface features. For a detailed explanation, please see the text.

Figure 4 shows the strike of more than 1700 linear features identified in the studied region. The lineaments show an NE–SW overall orientation, which is most likely strongly biased by the large number of lineaments belonging to the Wispy Terrain. Analyzing the groups belonging to various terrain units in the studied location, over 1200 linear features were found in Eurotas and Palatine Chasmata (Wispy Terrain), dominantly having a NE–SW strike orientation (Figure 4a). Almost 250 linear features were found at the "Prytanis–Alcander Plain", a terrain unit located east of the Wispy Terrain, named after two larger craters observed there. The WNW–ESE strike of those lineaments is quasi-perpendicular to the dominant direction of the lineaments of the Wispy Terrain (Figure 4a). Linea-

ments of Carthage Fossae (139 linea) have a strongly W–E-oriented cryotectonic system. Drepanum Chasma, with the smallest number of lines (88 linea) among the ones whose strikes were plotted, shows an N–S orientation. Only a few, 10 and 5, lineae were mapped, which may belong to Clusium Fossae and appear along Aufidus Catena, respectively. The former ten are located at the SE end of Clusium Fossae, overlaid by the lineaments of Eurotas Chasmata (Figure 4b). The latter, which appear along Aufidus Catena, may have some connection to the formation of the catena. Still, it is no more than speculation at this stage of the research.

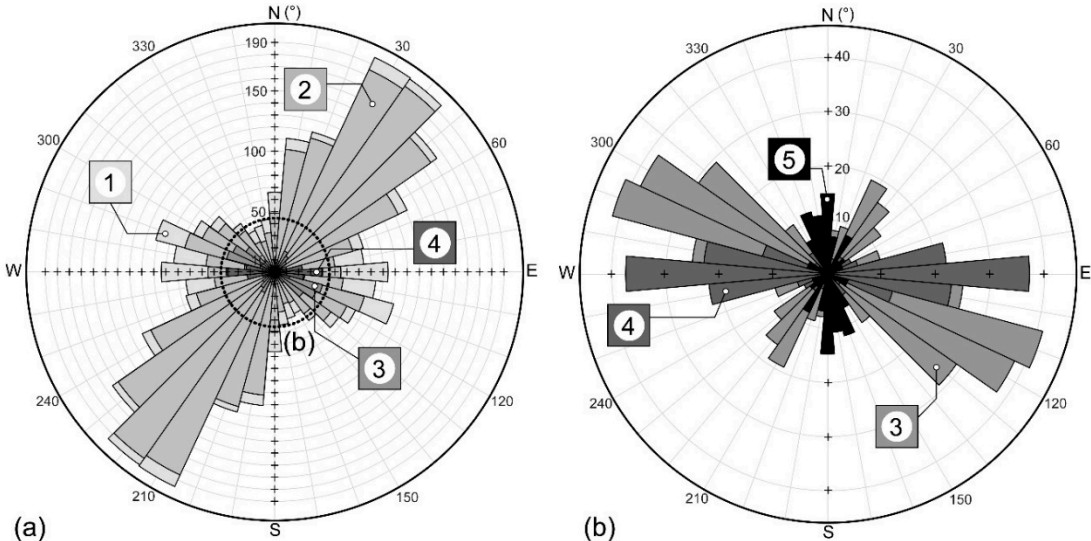

**Figure 4.** The strike of the linear features identified in the region (**a**) and a cut-out diagram with the ones with fewer linear features (**b**). The numbers indicate various linear feature groups belonging to different terrain units: 1—the whole studied location; 2—Eurotas and Palatine Chasmata (1222 linea); 3—the Prytanis–Alcander Plain (248 linea), a region eastward of Palatine Chasmata, named after two larger craters found there; 4—Carthage Fossae (139 linea); and 5—Drepanum Chasma (88 lines) (for more information about the referred terrain units, please see Figure 1b). The dotted circle shows the area of the cut-out diagram (**b**).

### 3.2. Tectonic Settings Observed on the Wall of Various Craters

Following the geological mapping of the area, the relationship between the tectonically active zones and the increasing number of incomplete Type TF craters appearing along those tectonic lines seemed clear (Figure 2a). As was already mentioned in Section 2.2, such crater types may indicate thrust faults.

To verify the results of the crater mapping about the possible appearance of thrust faults, the walls of larger craters, namely Silvius (D: 74 km), Daucus (D: 80 km), Amata (D: 76 km), and Catillus (D: 42 km), found in a region and crosscut by tectonic lines, were studied. Despite the potential bias due to the quality (at a low resolution) of the images, certain structural geological characteristics might be recognizable (Figure 5). Putative faults can be observed in the walls of larger craters, e.g., Silvius and Daucus (Figure 5a–d). They might be identified as domino-style normal faults. In addition, thrust fault-like features or, as an alternative to thrust faults, listric faults may be recognized "crossing" the Amata crater (Figure 5e,f). Unfortunately, it is hard to identify key features such as the thrust sheets of thrust faults, or alternatively, the "imbricating splays" of splay and décollement faults. The same applies to the putative listric normal faults, which may be the other possible candidates if the tilted face of the blocks is clearly observable. Such features may help to decide between the two possible fault types. Compared to the putative thrust faults, the features observed at the Catillus crater most likely indicate a normal fault system of a "graben" structure between two uplifted blocks (Figure 5g,h).

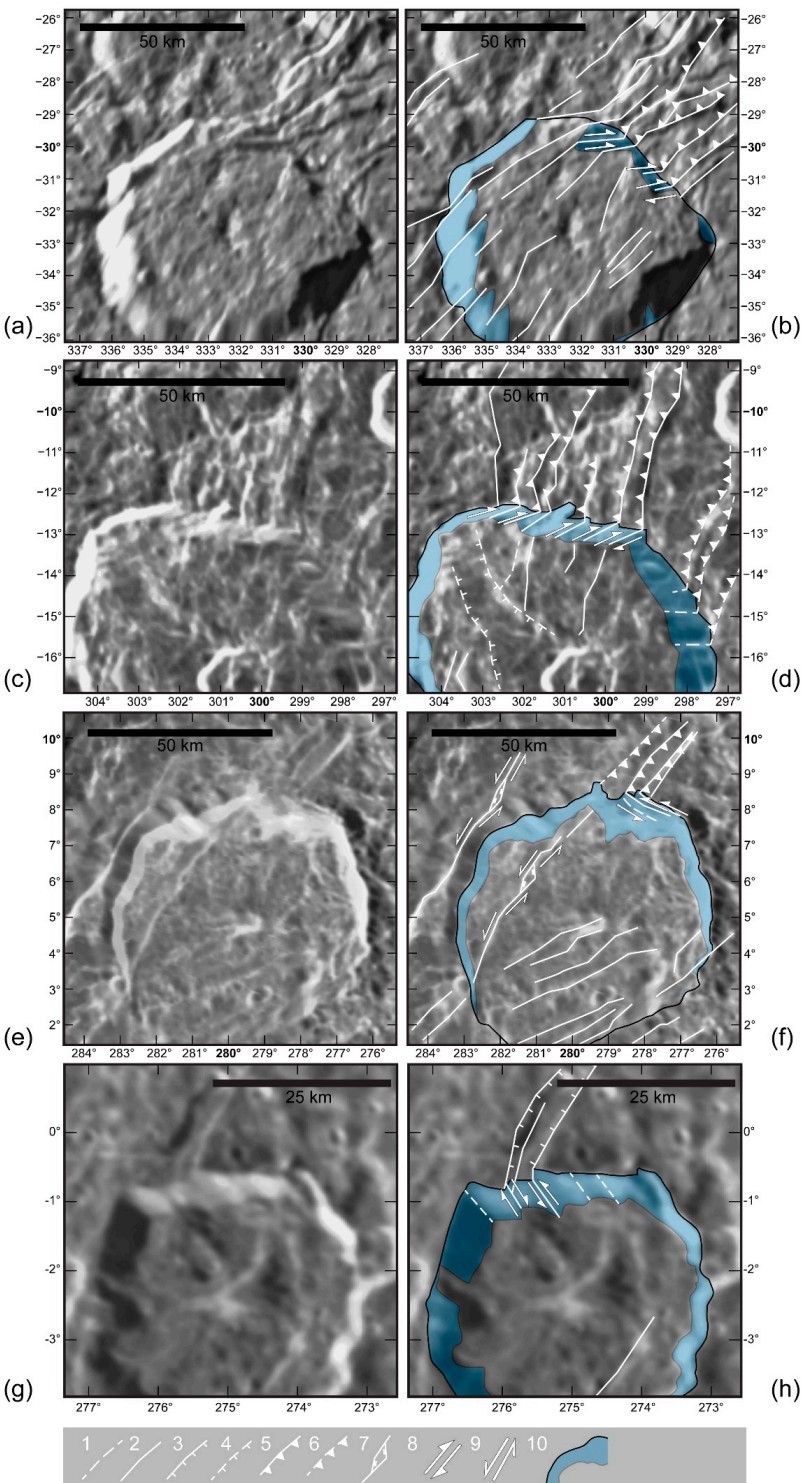

**Figure 5.** Structural geological characteristics observed in the walls of large craters crosscut by putative tectonic lines. (**a**,**b**) Silvius, (**c**,**d**) Daucus, (**e**,**f**) Amata, and (**g**,**h**) Catillus craters. The numbers in the legend represent the following features: 1—probable fault; 2—fault (uncertain direction of movement); 3—normal fault; 4—potential normal fault; 5—thrust fault; 6—probable thrust fault; 7—strike-slip fault with pull-apart basin; 8—orientation of (quasi-)vertical block movement; 9—direction of (quasi-)horizontal block movement; and 10—exposed crater wall. The figure is the modified version appeared in Bradák et al. [30], is reprinted/adapted with permission from Bradák et al. 2023 [30].

## 4. Discussion

### 4.1. Suspected Structural Geological Features on the Wispy Terrain

One key question in the studied area is the existence of thrust faults, which may change some general ideas about the tectonic processes appearing in the Wispy Terrain. Compared to the previous studies suggesting extensions in the region (summarized in Table 1; e.g., [8,9,15]), thrust faults would indicate a compressional stress field and convergence of the (ice) plates in general. In addition, thrust fault deformation may develop in a terrestrial environment when two blocks (hinterland and foreland) collide. It may lead to the formation of décollement and imbricate splays (so-called splay and décollement faults), one of the most characteristic components of subduction zones (e.g., the accretionary wedge) [40].

Type TF, fragmentary, or incomplete craters seem abundant along lineaments appearing in Eurotas and Palatine Chasmata. As is already described in Section 2.2, the formation of thrust faults would result in the "disappearance" of some part of the crater (by the hanging wall, which may "cover" a certain amount of the footwall top and the detached part of the crater as well), leaving only fragmentary craters behind. In such a case, the "bright" linear features would represent the faces of the hanging wall blocks exposed to sunlight. As an alternative to thrust faults, some features, e.g., the ones observed in the wall of the Amata crater, may be recognized as listric normal faults (Figure 5e,f). Due to the resolution of the image, it is hard to decide whether a thrust fault or a listric normal fault can be seen. Still, regardless of the nature of the observed fault, developing a listric normal fault would not result in the "disappearance" of half craters. As explained in Section 2.2, normal (or reverse) fault development would still allow the identification of a crater as a whole, even if the faulting deforms its shape.

Based on the investigated crater walls and neighborhood of the large craters, the putative faults observed in the walls of the Silvius and Daucus craters (Figure 5a–d) might be identified as domino-style normal faults [15,41] but supported by various evidence, such as the lack of a back-tilted face, the possibility of those features was already excluded [15].

As a summary, the complexity of Wispy Terrain's structural geological setting is shown by the appearance of a putative normal fault system, a "graben" structure between two uplifted blocks exposed in the crater wall of Catillus (Figure 5g,h). In addition to normal faults, indicating an extensional stress field and the divergence of ice blocks, the existence of putative thrust/splay and décollement faults suggests compression and the convergence of plates. The observation of the high abundance of tectonized polygonal craters and crater fragments in Wispy Terrain agrees with the statement of previous studies, suggesting the influence of a satellite-wide stress field and the presence of a widespread subsurface lineation [7,38].

### 4.2. Tectonic Settings on the Wispy Terrain

Based on the introduced evidence, further speculations about the characteristic tectonic settings at Wispy Terrain can be made, assuming that thrust faults exist in the region. Based on the allocation of the Type TF craters, the major part of Palatine Chasmata may be interpreted as a series of thrust faults, indicating a SE to NW compressional field and the possible movement of some ice blocks of the Prytanis–Alcander Plain in the same direction (Figure 6).

The a1 and a2 areas in Figure 6a indicate a characteristic setting that appeared at least two times in the studied region of the Wispy Terrain. It shows a set of SW to NE-oriented thrust faults associated with a dextral (a1; Figure 6a) and a sinistral (a2; Figure 6a) strike-slip fault system, indicated by Type SSF craters. A similar association of tectonic features can be found on Earth, e.g., in the Nankai subduction zone (Japan) and referred to as (mega-)splay fault system with wedge boundary strike-slip fault [42]. Applying their schematic model to the tectonic setting found in the Eurotas Chasmata (described above; Figure 6a, a1 and a2), the series of thrust faults can be defined as the "Outer wedge", which consists of imbricate thrusts; the "Transitional zone", which is represented by the strike-slip faults; and the

"Inner wedge" region, which can be located in the (boundary region with the) Intermediate Cratered Terrain, NW of the Eurotas Chasmata (Figure 7).

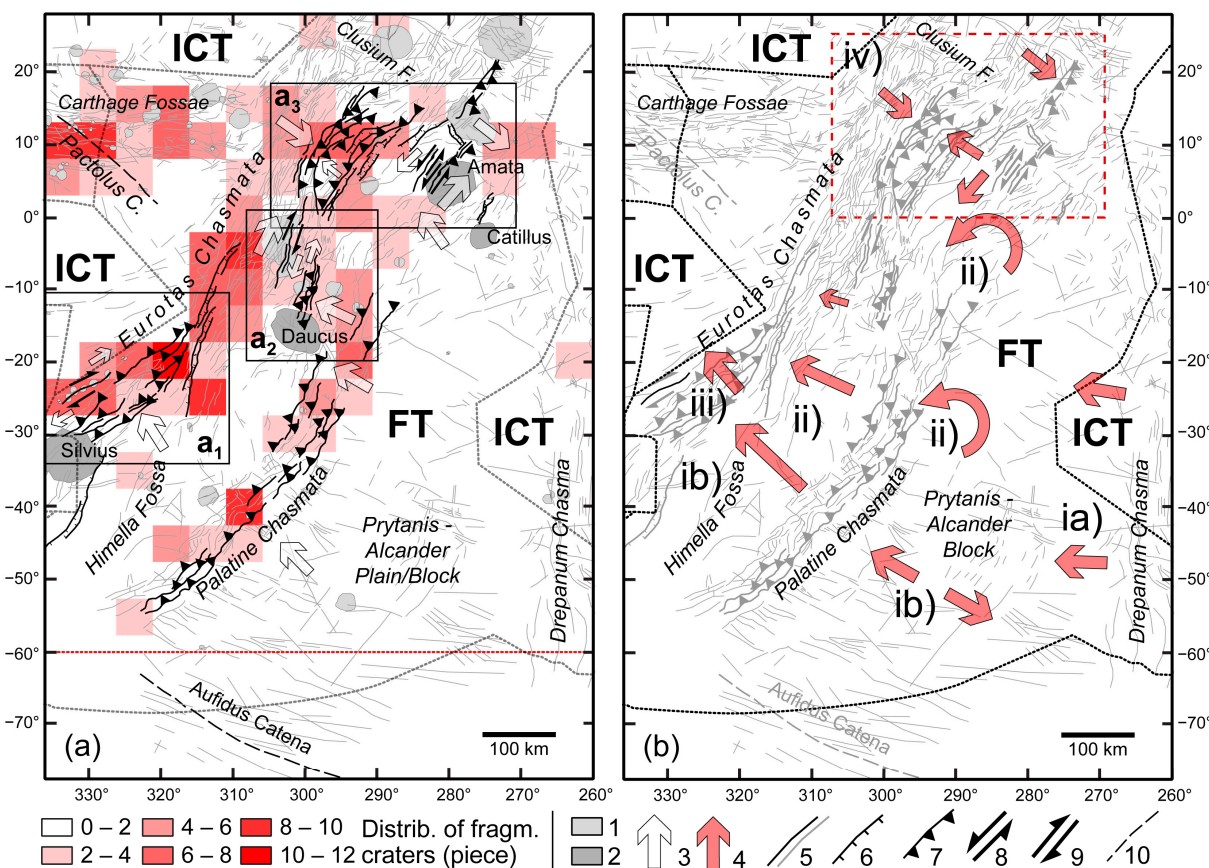

**Figure 6.** The relationship between various tectonic features is based on the distribution of various polygonal craters and crater fragments, the analysis of large crater wall "outcrops" (**a**), and the reconstruction of the tectonic development of the region (**b**). a1, a2, and a3 indicate locations with complex tectonic settings, characteristic of the studied part of the Wispy Terrain. The dotted lines represent the border between various terrains, named ICT (Intermediate Cratered Terrain) and FT (Faulted Terrain) [6]. The white-to-red gradual color background indicates the distribution of Type TF craters in the studied location, observed in a grid with a cell size of 50 km². Dashed red lines at latitude −60° indicate the region's boundary, with mapping uncertainties southward due to the projection-related distortion of the surface features. For a detailed explanation, please see the text. 1—Type P and SSF craters together; 2—craters with wall "outcrops" where tectonic features were studied; 3—indication of suspected tectonic movements; 4—reconstruction of main tectonic movements; 5—faults with uncertain direction of movements; 6—(probable) normal fault; 7—(potential) thrust fault; 8 and 9—orientation of quasi horizontal sinistral (left) and dextral (right) strike-slip fault movements, respectively; and 10—catenas. The marks (ia), (ib), (ii), (iii), and (iv) refer to the tectonic movements summarized in Section 5, shortly: (ia) extension triggered by the formation of Drepanum Chasmata; (ib) dilatation in Prytanis–Alcander block; (ii) closure of the area between Palatine and Eurotas Chasmata and the counterclockwise rotation of the Prytanis–Alcander block; (iii) putative subsumption between the Prytanis–Alcander Plain (block) and a block of Intermediate Cratered Terrain; and (iv) (the area inside the dashed red line) the renewal of some earlier tectonic lines of the Carthage and Clusium Fossae.

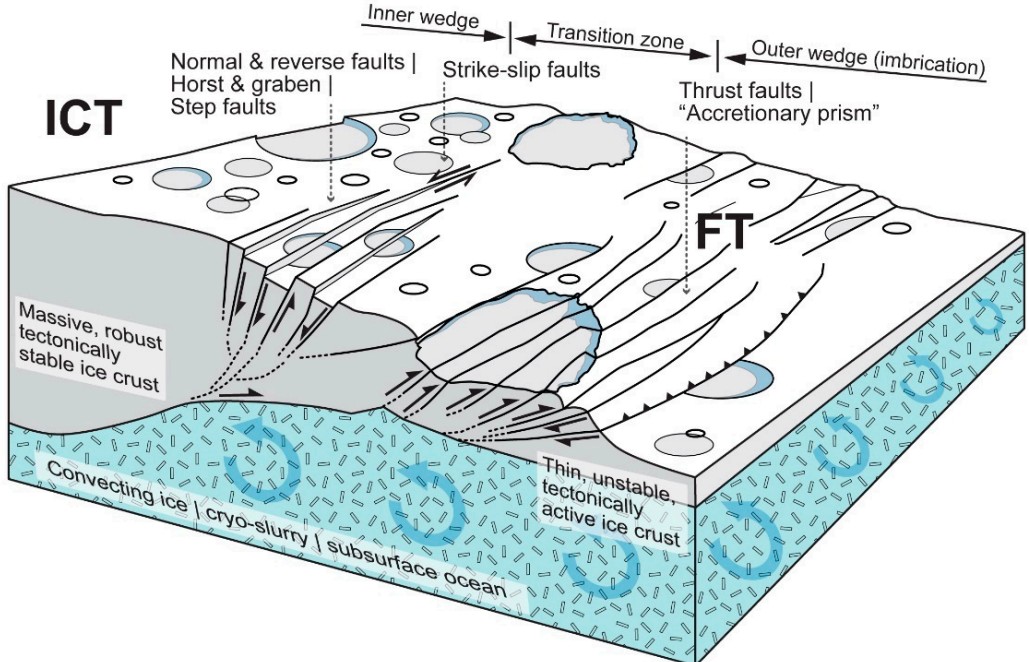

**Figure 7.** Model of a putative subduction-like (subsumption) setting on Dione, based on the tectonic features that appeared at the Wispy Terrain, observed at the Eurotas and Palatine Chasmata (Figure 6; a1 and 2). The reconstruction of various tectonic features is based on the schematic image of (mega-)splay fault system with wedge boundary strike-slip faults described and presented in Tsuji et al. [42]. The abbreviations ICT and FT refer to Intermediate Cratered Terrain and Faulted Terrain, respectively [6]. The figure is the modified version appeared in Bradák et al. [30], is reprinted/adapted with permission from Bradák et al. 2023 [30].

The complex tectonic setting in a3 location consists of three complex fault systems.

1. It is difficult to clearly describe the components of the bended-shape (SW to NE) set of normal and thrust faults appearing in the westward part of the area (Lat. 0–20°, Long. 300–280°; Figure 6a). They indicate a somewhat complex stress field system consisting of dilatation (characterized by the normal faults and horst and graben structure) accompanied by compression toward the neighboring E–SE-located regions (indicated by the putative thrust faults);
2. Some sets of the bent fault system were seemingly overlapped by the neighboring set of thrust faults, indicating roughly NW-directed compression toward the bent fault system (Figure 6a). This set of faults is located in the quasi-center part of the a3 area;
3. The third significant fault system is located in the eastern part of the a3 area. It consists of a set of thrust faults with SW to NE strikes and sinistral strike-slip faults (Figure 6a). The two groups of tectonic features are separated by an NW–SE-oriented trench, possibly indicating putative normal faults, partly overlaid by sets of perpendicular, parallel-running faults.

### 4.3. Speculation about Microplate Movements in the Studied Area

Extensional stress fields seem common on Dione's crust, developing various types of normal fault systems [15]. As shown in Table 1, the scale of the dilatational features stems from step faults, horst and graben structures to hemisphere-scale rift zones [8,9]. The discovery of potential thrust/splay and décollement faults may make the surface renewal models more complex than expected. The appearance of compression-related tectonic features, especially the ones whose morphology resembles the structure of an accretionary prism in a terrestrial environment, may suggest the existence of subsumption (subduction-like) processes on Dione [42] (Figure 7). Ideally, this means that many compo-

nents (phases) of a Wilson cycle-like tectonic cycle [43], namely rift formation, extension, collision of ice blocks (microplates), and subsumption (subduction), may appear in icy planetary bodies, and Dione's Wispy Terrain just turned out to be the model region for it. Based on such a new perspective, some speculation can be made about the region's development, considering the appearance of the "micro" Wilson cycle-related cryotectonic evolution of the ice plates (Figure 6b).

The geological processes appearing in the region seem to be one of the youngest in Dione, dating back from 2.7 Ga to 260 Ma [5], and the newest studies even suggest that the age of the studied faults on Dione's Wispy Terrain is only 152 Ma years old, implying that cryotectonism may be still active or was active a very short geological time ago (ca. 100 Ma) [12]. In addition, the two studied chasmata are located in the so-called Faulted Terrain, which is surrounded by the ice plates of older, so-called Intermediate Cratered Terrains [6], which may determine the direction of the cryotectonic evolution of the terrains.

Based on simulations [44] and Earth-based analogies [8,9], we may assume that pre-rift processes weakened specific parts of the Faulted Terrain and the Intermediate Cratered Terrain. An example of such a region would be the one marked "a3" (Figure 6a), with a complex cryotectonic setting introduced in Section 4.2, with some components indicating extension in the ice shell. The development of the extensional stress field in the ice shell and the formation of a rift system may be triggered by endogenic (e.g., phase change within the satellite, solid-state convection in the crust, and thermal plumes) and exogenic (e.g., diurnal tides, tidal forces, orbital forcing, and non-synchronous rotation of the ice shell) processes [8,9,45,46], and the references therein. Some sections of the ice shell in the area open and may spread due to continuous material accretion via cryovolcanic activity [12]. Later, the evolving and spreading microplate may collide with the thicker, more massive, stable, and possibly older terrain of the ice crust (Figure 6a), e.g., some evolving part of the Faulted Terrain may collide with the much older Intermediate Cratered Terrain [6]. This collision may lead to subduction-like processes, similar to the subsumption observed in the ice shell of Europa (Jupiter) [47] (Figure 7). Such collision and subsumption may result in the formation of the splay fault system with wedge boundary strike-slip faults, similar to the ones observed in subduction zones in a terrestrial environment [42].

## 5. Summary and Conclusions

Based on the described cryotectonic features, the region's development may be characterized by the following components and steps (Figures 6 and 7), which are in some ways similar to some phases of a Wilson cycle appearing in terrestrial plate tectonic systems.

- Putative rift formation phase and extension. The related tectonic features may refer to the first phase of a Wilson cycle, indicating extension and rifting mechanisms. The formation of the horst and graben system of Drepanum Chasma, one of the earliest among the chasmata, suggests intense extension at the eastern part of the studied area, between one of the microplates belonging to ICT and the Prytanis–Alcander Plain, a putative cryotectonic block or microplate. It may trigger the NW movement of the Prytanis–Alcander block. Such an extension may have some influence on the formation of Himella Fossa as well. These processes may be considered the phase of rift formation, the juvenile stage of microplate formation in a Wilson cycle;

- Extensional processes in a back-arc basin-like tectonic setting. As an alternative to the previously described phase, the NW movement of the Prytanis–Alcander block or microplate is triggered by the pulling effect of the thrust fault system of Palatine Chasmata. This process is indicated by many lineaments parallel and perpendicular to Palatine Chasmata and might result in the extension and renewal of the horst and graben system of Drepanum Chasma. Such a tectonic setting may be analogous to a back-arc basin-like setting on Earth, characterized by extension, and may function as a precursor for rifting processes.

- Phase of early microplate collision. The closure of the area between Palatine and Eurotas Chasmata. The decreasing distance between the main thrust fault sets of

the chasmata from the SW section toward the NE region of the studied area shows such closure. It may suggest the counterclockwise rotation of the Prytanis–Alcander block, resulting in such closure between the Eurotas and Palatine Chasmata. These processes may be considered a mature phase in microplate tectonism, as indicated by the collision of various blocks;

- Putative phase of subsumption (subduction-like process on icy satellites). The formation of a splay fault system with wedge boundary strike-slip faults and the collision and appearance of putative subsumption between the Prytanis–Alcander plain (block) and a block of the Intermediate Cratered Terrain, located NW of the plain. This process may indicate the senile phase of the Wilson cycle, ending with the disappearance of some microplates, marking the end of their subsumption;
- The possible influence of tectonic features developed in an earlier cryotectonic phase, such as Carthage and Clusium Fossae formation. The possibly overlapping section of the fault system of those older features and the younger Eurotas (and Palatine) Chasmata may have resulted in the renewal of some earlier tectonic lines.

Despite its sometimes speculative nature, the "re-investigation" of the Wispy Terrain provided some new information that may be considered in future investigations, targeting one of the most enigmatic regions of Dione. The results of geological mapping of the Wispy Terrain, indicating the possibility of Wilson cycle-like tectonic cycles on icy satellites, may provide the foundation for surface modeling related to ice physics as well as the modeling of the moon's interior, e.g., in response to orbital forcing and tidal heating.

**Author Contributions:** Conceptualization, B.B.; methodology, B.B.; formal analysis, B.B. and J.K.; writing—original draft preparation, B.B.; writing—review and editing, J.K., D.A., M.E.Y. and C.O.; visualization, B.B. All authors have read and agreed to the published version of the manuscript.

**Funding:** This research received no external funding.

**Data Availability Statement:** Data related to this research will be sent upon request.

**Acknowledgments:** We want to thank the reviewers of this article for taking the time and effort to review the manuscript. We appreciate their valuable comments and suggestions, which helped improve this manuscript's quality.

**Conflicts of Interest:** The authors declare no conflict of interest.

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
