# Peer review of "Introduction to Dione’s Wispy Terrain as a Putative Model Region for “Micro” Wilson Cycles on Icy Satellites"

_remotesensing, doi:10.3390/rs15215177_

Round 1

Reviewer 1 Report

Comments and Suggestions for Authors

A nicely researched and written manuscript of great interest in the planetary sciences/planetary surfaces field. I found only a few minor faults and have a few suggestions. Overall recommendation: Accept with only minor text editing.

Line 159: The url for the Dione image has a misplaced dash in mo-saic. When you click on the link it goes to an Error 404 page. Remove the dash and it works just fine.

Line 410: Do not start a sentence with a lower case letter. Instead of "a1 and a2...." perhaps "the a1 and a2....."

In Figure 1- left blow up I suggest that the 4 craters shown in Figure 5 are denoted in Red and that the caption states something like (Red crater analysis shown in Figure 5).

Finally the issue of uncertainty in mapping due to the low resolution is not discussed anywhere. There needs to be a brief sentence or two on what the poor resolution means for mapping uncertainty, how it might impact the interpretations and how future missions might be designed.

Reviewer 2 Report

Comments and Suggestions for Authors

Bradak et al submitted their manuscript covering Dione's Wispy Terrain for review and potential publication in remote sensing.
In their study, the authors investigate Dione's poorly understood assembly of tectonic surface features. The authors' geologic and structural analyses of a larger surface region allow them to link this terrain and its inventory to specific cryotectonic processes and to the proposal of a larger-scale governing tectonic mechanism.

The text is well-written; figures and charts are well-crafted and of outstanding quality. Figures and tables complement the analyses and descriptions and are considered much needed to understand the overall story. It should be noted that some of the figures appear in a proceedings volume, published by the same publisher. Also, significant text fragments have been taken from that contribution. For several paragraphs, this ends in verbatim copies of larger chunks of text to a degree which makes me feel uncomfortable. After all, the similarity index is at 30% and that affects not only the introduction and methods but also the discussion and results. The authors might want to check for these issues.

Some comments regarding contents:

The study is highly speculative and the authors do not hide that. After all, there are just a few images, and no decent chronological models or usable geophysical models (to my knowledge). To me, it is important that the area of interest has been analyzed in detail with respect to an overarching objective, that the work has been conducted with care, and that the implications are communicated as being either likely or speculative. The connection to a Wilson cycle might be bold but it is valid in a way that it is based on observed features. Much of the geophysical modelling for planetary surfaces is based on such observations and a number of geophysical studies on volcanic developments were based on detailed structural mapping. I would strongly suggest not to cut off this investigation where it stands now, with some mentioning of its speculative character. I would rather suggest establishing a link to the potential of this investigation for future modelling that could be based on such mapping. This could involve surface modelling related to ice physics, but also to the modelling of the moon's interior in response to its orbit geometry.

The introduction is well-written and establishes the overall context well.
It should be noted that much of the introduction has been taken verbatim from a proceedings paper published by the first author. I am aware it's not related to the analysis but I feel a bit uncomfortable here and would suggest some re-wording.

L43: the authors might want to consider using "planetary geologic mapping" rather than "astrogeologic mapping" perhaps?  Not much "astro" here.

L142: "The goal of this study to identify various cryotectonic features and stress fields by a crater-morphology based method, and reconstruct some phase of Wispy Terrain's tectonic  evolution, one of the most, and still enigmatic region of Dione." --
These features are essentially tectonic and not purely cryotectonic, is that correct? Would you be able to identify processes that you would assign to cryospheres only? The reason why I am asking is that the reader might expect cryotectonic features and processes although you refer to tectonic features related to the icy crust of Dione. There is some difference in that. I would ask the authors to perhaps introduce a short sentence mentioning this unless I misunderstood the approach.

Check Figure 1 references. Format does not correspond to MDPI's formatting and the spelling of authors is "GreelEy", as well as "Schenk".

L168: why mention what the authors did not do? As long as they do not claim machine learning has been used everyone assumes it was digitized manually,  If the effort should still be highlighted I would suggest wording such as: "contacts and structural elements were interpreted visually and digitized interactively " or something similar.

L190: "Polygonal craters and crater fragments are craters with their floor or wall cut by faults, which might cause the detachment of certain parts and their vertical or horizontal allocation." -- unclear, could that be explained in more detail? The results give some insights but it remains unclear here.

L235: "were catalogized" -- categorized?

L236: "ii) Type P – polygonal craters formed by various cryotectonic processes (Fig. 2a)," -- how were these various cryotectonic processes identified? And what are those exactly? See the above comment.

The results are fine by me, however, there are some elements that could be seen as discussion, I would ask the authors to re-check to keep the results clean from speculation.

L484: For the summary/conclusions the authors might want to think about a depiction of the cycle with respect to Dione's observed features. The description reads well, but given the larger-scale implications, the summary focuses too much on specific regions and observations (in my opinion). Thus, the take-away message is hard to find. I believe the story could be sold better by showing the stages in relation to observations.

Also, as mentioned above, I would strongly suggest including some perspective, i.e. how these observations/mapping could be used for future modelling and in particular testing.

Comments on the Quality of English Language

English is in general fine. Just some minor grammar issues and occasionally wordy sentences. Can likely be fixed during copy-editing.

Reviewer 3 Report

Comments and Suggestions for Authors

Saturn’s icy moon Dione and its enigmatic feature have been the target of a long-lasting scientific debate. In this study, the authors provide a detailed geological and cryotectonic analysis in the surroundings of the Eurotas and Palatine Chasmata and proposes additional unidentified cryotectonic processes and a formation model. 

The authors gave a full understanding of the crater types from the geologic mechanism. The results are helpful to further understand the icy moon including Dione.

My only comment is that the arrangement of the manuscript should be improved. E.g., Section 1 only has one subsection.
